# Porous Polymers Based on 9,10-Bis(methacryloyloxymethyl)anthracene—Towards Synthesis and Characterization

**DOI:** 10.3390/ma16072610

**Published:** 2023-03-25

**Authors:** Małgorzata Maciejewska, Mateusz Józwicki

**Affiliations:** Department of Polymer Chemistry, Institute of Chemical Sciences, Faculty of Chemistry, Maria Curie-Skłodowska University in Lublin, Gliniana 33, 20-614 Lublin, Poland

**Keywords:** polymeric microspheres, porous structure, thermal resistance

## Abstract

Porous materials can be found in numerous essential applications. They are of particular interest when, in addition to their porosity, they have other advantageous properties such as thermal stability or chemical diversity. The main aim of this study was to synthesize the porous copolymers of 9,10-bis(methacryloyloxymethyl)anthracene (BMA) with three different co-monomers divinylbenzene (DVB), ethylene glycol dimethacrylate (EGDMA) and trimethylpropane trimethacrylate (TRIM). They were synthesized via suspension polymerization using chlorobenzene and toluene served as porogenic solvents. For the characterization of the synthesized copolymers ATR-FTIR spectroscopy, a low-temperature nitrogen adsorption–desorption method, thermogravimetry, scanning electron microscopy, inverse gas chromatography and size distribution analysis were successfully employed. It was found that depending on the used co-monomer and the type of porogen regular polymeric microspheres with a specific surface area in the range of 134–472 m^2^/g can be effectively synthesized. The presence of miscellaneous functional groups promotes divergent types of interactions Moreover, all of the copolymers show a good thermal stability up to 307 °C. What is important, thanks to application of anthracene derivatives as the functional monomer, the synthesized materials show fluorescence under UV radiation. The obtained microspheres can be used in various adsorption techniques as well as precursor for thermally resistant fluorescent sensors.

## 1. Introduction

Porous materials play a crucial role in a modern society. A variety of contemporary achievements relies on porous components. These materials are successfully utilized as column packing in different chromatography techniques [1,2,3,4,5,6,7,8,9], as filtration/separation membranes [10,11], as support for sensors and catalysts [12,13,14], as transporters in drug delivery systems [15,16,17]. They are also widely applied for detection and removal of pollutants from water [18,19], for CO_2_ capture [20,21], separation of rare earth elements [22] and many others advanced applications [23,24,25,26,27,28,29]. Porous materials can be obtained either by creating small particles or clusters where the surface-to-volume ratio of each particle is high, or by producing materials where the void surface area (pores) is high compared to the number of particles. These pores can be created through different methods, such as using a porogen during polymerization, or by post-processing the polymer. As a result, an amazing variety of porous products is constantly created and implemented. To improve trucking down and analysis of this impressive class of materials, various types of classifications are applied by scientists around the world. One of the commonly accepted division includes three main groups. Firstly, natural materials with high surface area. There are several natural materials that have a strongly developed internal surface such as, zeolites, clays, pumice, peat, metal oxides or activated charcoal. Zeolites are naturally occurring minerals that have a high surface area and a porous structure. They are composed of aluminum, silicon, and oxygen, and are characterized by a three-dimensional network of tetrahedrally coordinated aluminum and silicon atoms. The pores in zeolites are typically in the range of 0.3 to 1 nanometer in diameter. They are widely used in catalysis, organic synthesis, solar energy storage and use, gas separation and other chemical and biomedical application [30,31]. In the case of clays, their structure significantly varies as a result of the function of the mineralogical reserve. What is important, on the basis of clay minerals, more advanced porous clay heterostructures (PCH) can be easily obtained. PCH materials were studies as catalysts of various reactions, as adsorbent for CO_2_ capture, removal of heavy metal and dyes [32,33,34,35,36]. In turn, peat is a soil-like material that is composed of partially decomposed plant material. Peat has a high surface area, a relatively high water-holding capacity, and a spongy texture. For dyes and metals in wastewater, it can be used as an adsorbent [37]. However, the oldest and still extensively utilized natural adsorbents are porous carbons. They have been among the most explored materials due to their low cost, good stability, and high specific surface areas [38,39,40]. The second significant group are metal–organic frameworks (MOFs). These structures are made up of organic linkers that are held together by metal atoms. As a result, they look like a molecular cage. The metal ions or clusters provide the framework, while the organic ligands provide the porosity. This combination of metal ions or clusters and organic ligands creates a highly porous structure with a large surface area. They are used in desalination, carbon capture and storage, gas separation, biological imaging and sensing and play many other essential function [41,42,43,44,45].

Finally, permanently porous polymers. These highly cross-linked, three-dimensional networks possess superior inherent porosity, excellent stability, pre-designable and tunable structures. They are a broad class of advanced materials that have gained remarkable attention [46,47,48,49,50]. Porous polymers are mainly obtained during the radical copolymerization with cross-linking with the presence of diluents. At a certain degree of conversion, the homogeneous polymerization mixture is separated into solid and liquid phases: The solid phase is made up of a cross-linked polymer network, whereas of the solvent and unreacted comonomers consist liquid phase. The process of phase separation occurs. It is determined by the increase in cross-linking degree (*ν*-syneresis) or the alternation interactions of polymer-solvent (*χ*-syneresis). In the second case, it can be macro- or microsyneresis. In the case of microsyneresis, the liquid phase is dispersed in the continuous phase of polymer gel. During macrosyneresis, the nuclei aggregated. They are microgels of higher a degree of cross-linking surrounded by a liquid phase. The resulting microgels react with each other by free vinyl bonds to form agglomerates (macrogels). The free volume between agglomerates creates macropores that is, according to the IUPAC recommendation [51], pores larger than 50 nm in diameters. Mesopores with pore size in the range of 2−50 nm can be found between macrogels and micropores (linear dimension below 2 nm) are present between microgels. Generally, porous polymers display micro-, meso- and macroporosity, which supports diversified application. The high microporosity of polymers allows them to be used in the adsorption and separation of small hydrocarbons or CO_2_. The mesoporosity make them excellent candidates for catalysts or catalytic supports due to their high specific surface area and high stability. Additionally, the mesoporosity also support the adsorption of big molecules, such as aromatic compounds. Due to the presence of macroporosity, the polymers have a high potential for the adsorption of large molecules such as dyestuff or complex biomolecules.

A significant variety of the polymer implementation creates a need for materials which possesses distinct or even conflicting properties. In our approach, we targeted into polymeric microspheres with considerably developed porous structure, diversified functional group and high thermal stability. As indicated in our previous study [52] synthesis of porous copolymers based on aromatic divinylbenzene (DVB) with different comonomers leads to materials with a good thermal resistance. For all synthesized materials, the *T*_5%_ determined in helium atmosphere was above 300 °C: Following this strategy, we used a tetrafunctional methacrylate monomer based on anthracene, i.e., 9,10-bis(methacryloyloxymethyl)anthracene (BMA). BMA is a methacrylic derivative of anthracene, an important representative of polycyclic aromatic hydrocarbons (PAH). Anthracene contains three fused benzene rings. The aromatic rings that form the monomer structure provide the π-π interactions. Anthracene is colorless but exhibits fluorescence under ultraviolet radiation so it can be easily used as a UV tracer, e.g., in the conformal coatings applied in printed wiring boards [53]. Similar behavior can be observed for other polycyclic aromatic hydrocarbons. The simplest PAH is naphthalene. Its derivative, i.e., naphthalene-2,7-diol was used to synthesis the reactive monomer 2,7-di(methacyloyloxy)naphthalene. After excitation by UV radiation, it emits yellow radiation. It was successfully applied in the copolymerization with 2-hydroxyethyl methacrylate, N-vinyl-2-pyrrolidone, styrene, 1,4-divinylbenzene and methyl methacrylate [54]. Additionally, the synthesis of aromatic thiomethacrylate derivatives of naphthalene and their copolymers was reported [55]. It was of interest to test the possibility of BMA application in synthesis of fluorescencent porous copolymers in the form of microspheres. The proper selection of the polymerization technique allows to obtain microspheres with a wide spectrum of diameters. The main factor that determines the choice of the suitable method is the potential application. In the case of some essential techniques (e.g., PALS studies, gas chromatography, catalysis), the required size of the microspheres is about 100 µm. In order to achieve the desired diameter of the porous copolymers, the suspension technique is commonly employed. The suspension polymerization takes place under the influence of an initiator dissolved in the monomers dispersed in an aqueous solution stabilized by a proper agent. The type and amount of the stabilizers determines the size and quality of the microspheres. Based on our earlier studies [56,57], it was found out that polyvinyl alcohol of molecular weight 72,000 is a stabilizer that allows to obtain polymer microspheres having the desired diameter and perfectly spherical shape. However, the applied amount (6.5 g/195 mL H_2_O) generated some problems with the effective removal of the stabilizer. In this study, we decided to investigate the influence of stabilizer concentration on the quality of the synthesized microspheres. The main goal was to achieve regular, porous, highly crosslinked microspheres based on 9,10-bis(methacryloyloxymethyl)anthracene (BMA).

As co-monomers divinylbenzene (DVB), ethylene glycol dimethacrylate (EGDMA), and trimetylpropane trimethacrylate (TRIM) were applied. Chlorobenzene and toluene served as porogenic solvents. The obtained copolymers were thoroughly characterized by ATR-FTIR spectroscopy, thermogravimetry, scanning electron microscopy, a low-temperature nitrogen adsorption–desorption method, and size distribution analysis.

## 2. Materials and Methods

### 2.1. Chemicals

TRIM from Sigma-Aldrich (Schnelldorf, Germany), DVB and EGDMA from Merck (Darmstadt, Germany) were washed with 5% aqueous sodium hydroxide to remove inhibitors. Poly(vinyl alcohol) (PVA, MW 72,000) and 2,2′-azoisobutyronitrile (AIBN) achieved from Fluka (Buchs, Switzerland) were used as received. Reagent grade acetone, toluene, chlorobenzene, benzene, butan-1-ol, methanol, pentan-2-one, and sodium hydroxide were purchased from POCh (Gliwice, Poland). BMA was obtained in our laboratory according to the procedures described earlier [58].

### 2.2. Microspheres Synthesis

In our previous studies [52], we used quite a high amount of PVA to prevent of agglomeration of the synthesized microspheres and obtain a required diameter of the final beads. However, this approach led to some problems with the entire removal of the stabilizer from the porous surface of the product. Consequently, in the presented study, we conducted preliminary research focused on well-adjusted quantity of the stabilizer. The weight percentage of PVA in water was increased from 0.1 to 2.5 wt.%. It came out that application of a low PVA concentration resulted in agglomerated microspheres (Figure 1a). In the case of BMA copolymers, 1.5 wt.% of PVA turned to be sufficient for protection of the resulting microspheres and therefore this amount of stabilizer was used in further synthesis. It was stirred with water in a three-necked flask for 6 h at 80 °C. After dissolving PVA, the organic solution of 15 g of the monomers (BMA with addition of a given comonomer), 0.075 g of an initiator (AIBN), and 22.5 mL of the diluent (toluene or chlorobenzene) was made and slowly put into the stirred aqueous medium. The molar ratio of DMA to a comonomer was kept 1:1 in the first series of synthesis; in the second, the molar ratio was 1:4. In order to obtain copolymers with highly developed internal structure, toluene and chlorobenzene were used as pore-forming diluents. The copolymerization process continued for 20 h at 80 °C. Porous microspheres obtained in this process were collected, filtered, washed with hot water and subjected to an intensive cleaning procedure described by Tuncel [59]. However, even this careful cleaning was not sufficient for complete removal of the stabilizer (Figure 1b). The purified copolymers were dried in a vacuum exsiccator at 60 °C. Table 1 delivers the designations of the obtained copolymers along with the main experimental parameters.

### 2.3. Measurement Methods

Structural characterization of the obtained materials was made on the basis of FTIR spectroscopy. For this purpose, a Bruker Tensor 27 FTIR spectrometer (Ettlingen, Germany) operating in the spectral range 4000–600 cm^−1^ was used. The ATR-FTIR spectra were recorded with a resolution of 4 cm^−1^.

Porous structure of the investigated copolymers was ascertain based on the nitrogen amount adsorbed on the copolymers surface. The isotherms of nitrogen adsorption/desorption were collected at −196 °C using ASAP 2405N analyzer (Micromeritics Corp., Norcross, GA, USA). Before the porous structure measurements, the copolymers were outgassed (10^−2^ mm Hg) at 140 °C for 2 h. The linear Brunauer–Emmet–Teller (BET) plots were applied to calculate the specific surface area (*S_BET_*). The pore volume (*V*) was evaluated based on a single point adsorption at a relative pressure *p/po* = 0.985. Pore size distributions (PSD) and its maxima (PSD_max_) was appraised according to the Barrett, Joyner and Halenda (BJH) approach [60]. The pore diameters (*D_BJH_*) were evaluated on the basis of the PSD_max_.

A scanning electron microscope (SEM) Duall Beam^TM^, Quanta3D FEG (Fei Company, Hillsboro, OR, USA) was used for taking the images of the obtained microspheres. Their size and size distribution were measured by a Mastersizer Analyser 2000 (Malvern, Instruments Ltd., Worcestershire, UK). The British standard BS2955:1993 was the base for calculation of the distribution statistics. According to the standard, *D*(0.1) is the particle size below which 10% of the sample lies, whereas *D*(0.9) is the particle size below which 90% of the sample lies. *D*(0.5) refers to Mass Median Diameters (MMD) that is the size at which 50% of the sample is smaller and 50% is larger. All of the particle diameters are given in microns. Span (width of the size distribution) was calculated on the basis of the following formula:(1)$$span=\frac{D(0.9)-D(0.1)}{D(0.5)}$$

Thermogravimetric (TG) analyses were performed using a Netzsch STA 449 F1 Jupiter thermal analyzer (Selb, Germany) both in helium as well as synthetic air atmosphere (flow = 20 mL/min). The analysis temperature range was 35–800 °C whereas the heating rate was 10 °C min^−1^. The measurements were performed in Al_2_O_3_ crucibles. As a reference, an empty Al_2_O_3_ (mass of ~160 mg) crucible was used. The sample masses were about 10 mg. On the basis of the TG curves, the characteristic temperatures of *T*_5%_, *T*_20%_, *T*_50%_ mass losses as well as final decomposition temperature (*FDT*) were determined. In turn, on the basis of differential TG (DTG) curves, the temperature of the maximum rate of mass loss (*T*_max_) for individual decomposition stages was gauged.

Inverse gas chromatography (IGC) was applied for the determination of the polarity of the copolymers under study. To determine this parameters McReynolds procedure [55] was put to use. This method based on McReynolds’ constants (Δ*I*) of test substances: benzene (x), butan-1-ol (y), pentan-2-one (z). The constants can be determined by computing the difference between the Kovats’ index for benzene, butan-1-ol, pentan-2-one on the investigated stationary phase and graphitized thermally carbon black (GTCB). The overall polarity of the phase under study can be established by the sum of the McReynolds constant. The chromatographic measurements were performed Dani GC 1000 gas chromatograph (Milan, Italy) fitted with a thermal conductivity detector (TCD), an injector, and stainless-steel columns (100 cm × 1.6 mm I.D.) packed with the studied porous microspheres. The retention times for the test compounds were determined in helium at a flow-rate of 50 mL/min. A 1 µL syringe (SGE, North Melbourne, Australia) was used for manual injection of the samples. The measurements temperature was kept isothermally at 140 °C.

## 3. Results

In the synthesis of permanently porous microspheres, a suspension polymerization technique was used. Compared to previous studies, we have significantly reduced the amount of the used PVA, still obtaining regular microspheres. In their synthesis, three commercially available monomers were copolymerized with 9,10-bis(methacryloyloxymethyl)anthracene. The chemical structures of the monomers used in the synthesis of porous copolymers are displayed in Figure 2.

To settle the assumed chemical structure of the obtained microspheres, ATR-FTIR spectrophotometry was applied. The obtained spectra of the copolymers under study are displayed on Figure 3. As the copolymers exhibit some analogous functional groups, bands pointing the presence of aromatic rings, i.e., at 3066–3019 cm^−1^ (*ν* C-H), 1603 cm^−1^ and 1458 cm^−1^ (*ν* C-C), ester groups at 1726 (ν C=O) cm^−1^ and 1249–1139 cm^−1^ (ν C-O) as well as alkyl and alkylene groups, i.e., at 2962–2830 (*ν* asym. and sym. C-H) are visible on each spectrum. A weak absorption band at 1638–1630 cm^−1^ connected with the unreacted C=C double bonds can be also seen in some spectra. In the case of BMA-*co*-DVB copolymers, bands at 755 and 708 cm^−1^ (δ oop C-H of monosubstituted benzene) and at 829–796 (δ out-of-plane C-H of *p*-disubstituted benzene) are also visible.

The premediated objective of this research was to synthesize new polymeric materials with highly developed internal structure. In order to obtain this goal, BMA with different methacrylic and vinyl comonomers was polymerized in the presence of two different solvents. All of the used comonomers possess a high functionality, which leads to the high degree of crosslinking. The functionality of EGDMA and DVB is the same as that of BMA and is equal to four. It is even higher and equals six in the case of TRIM. This fact creates an excellent environment for *ν*-syneresis.

As evidenced in Table 2, the increase in monomer functionality expands the porous structure. BMA-*co*-TRIM copolymers have the highly developed internal structure. This phenomenon is reflected by high value of specific surface area, pore volume and diversified pore size distribution. What is important, not only is the type of comonomer important for the process of porous structure formation, but also its amount. In the first turn, the copolymers were synthesized using equimolar rations. Afterwards, the amount of the comonomer (EGDMA, DVB or TRIM) was increased and the molar ratio of BMA to comonomer was equal to 1:4. The increase in the comonomer amount in the polymerization medium gave rise to creation of a huge number of low energy and highly crosslinked nuclei during the phase separation. After the nuclei aggregation, the resulting microgels are very complex and create highly developed internal structure. The value of the specific surface area of copolymers obtained with the comonomer excess is considerably higher compared with their equimolar counterparts. A similar relationship can be observed with the pore volume. The chemical structure of the comonomer also has a great impact on the pore diameter. Figure 4 displays pore size distribution of microspheres obtained from equimolar amount of comonomers with toluene as porogenic solvent. As can be seen, the pore diameters strongly depend on the used comonomer. BMA-*co*-DVB copolymers are characterized by narrow pore size distribution and pore diameters approaching the micropore region. In the case of BMA-*co*-TRIM materials, the distribution is wider and apart from a small peak at 3.5 nm the maximum pore diameter is observed at 15 nm. The copolymers obtained on the basis of EGDMA possesses larger pores. Their size distribution is shifted towards macropores, and the maximum can be seen at 40 nm. The same pattern can be noted for the copolymers with molar ratio of comonomers equals to 1:4. The pores are slightly smaller, but the described tendency is preserved. What is important, the reduced diameter of the pores contributes to more developed internal structure (Table 2). What is more, the application of chlorobenzene in the synthesis of porous copolymers gave rise to further increases in the main parameters characterizing the porous structure. This phenomenon is connected with the better solvation power of chlorobenzene compared with toluene. Its Hildebrand solubility parameter is equal to 19.6 (MPa)^0.5^ whereas this of toluene is 18.2 (MPa)^0.5^.

Generally, the formation of porous materials was possible thanks to the application of the suspension technique in the synthesis. The characteristic feature of the suspension technique is the fact that the size and size distribution of the obtained microspheres depends on many polymerization factors (the diameter of a stirrer and a flask, the volume ratio of the water phase to the organic phase, the mixing power and the stabilizer concentration) [61]. Due to the chemical diversity of monomers, differences in organic phase densities were observed, and consequently, different interfacial tension was present. This phenomenon was evident in the synthesized microspheres size. The mass median diameter of BMA-*co*-DVB copolymers did not exceed 100 µm. The use of methacrylate comonomers results in microspheres of larger size. The diameters of BMA-*co*-EGDMA copolymers are in the range 89–184 µm, whereas BMA-*co*-TRIM copolymers in the range 76–217 µm depending on the molar ratio of the monomers and the used solvent. The detailed data of particle size and particle size distribution are collected in Table 3.

The referred differences are also clearly visible in Figure 5. The SEM images show the regular shape of the synthesized microspheres and confirm the data obtained using laser diffraction.

One of the most important characteristics of polymeric materials is their thermal resistance. In this study, thermal properties of the investigated copolymers were evaluated on the basis of thermogravimetry. Based on TG studies, it can be concluded that all of the copolymer under study exhibit good thermal stability in inert as well as oxidative atmospheres (Table 4 and Table 5). Among them, the best thermal resistance demonstrated BMA-*co*-TRIM copolymers. Its initial decomposition temperature was above 300 °C. This phenomenon is caused by the high (equal six) functionality of TRIM and its possibility to form highly crosslinked polymeric network. On the other hand, the value of final decomposition temperature was observed for BMA-*co*-DVB copolymers. The chemical structure of the DVB monomer can explain this fact. It possesses an aromatic ring that contributes to better thermal stability. Increasing the molar ratio to 1:4 for DVB as well as for TRIM resulted in the enhancement of thermal resistance of the synthesized copolymers. The opposite relationship was observed for EGDMA. Generally, the least thermally stable are copolymers of this tetrafunctional, aliphatic comonomer. All the indicators of thermal resistance determined for BMA-*co*-EGDMA microspheres have the lowest value among all investigated materials.

What is worth noticing, thermal decomposition in helium atmosphere of all investigated copolymers proceeded in three, partially superimposed steps (Figure 6).

In the case of synthetic air, all the main indicators of thermal resistance possess lower values compared with their counterparts determined in helium. The oxidative atmosphere accelerates the thermal decomposition of the investigated microspheres and the went in three main stages (Figure 7) with the maxima from 302 to 461 °C. The noises that happened in the plots are associated with the high sensitivity of the thermal analyzer. They occurred at the end of measurements and did not have relevant impact on the acquired results.

What is interesting, the type of porogen used in the synthesis did not have any impact on the thermal behavior. The copolymers obtained in the presence of chlorobenzene have almost identical value of characteristic temperatures as in the case of toluene.

The application of various monomers in the synthesis of polymeric microbeads results in different chemical characters of the surface. One of the suitable methods for investigation of surface character is inverse gas chromatography. Microspheres for gas chromatography should be thermally stable, quite rigid, and have a large interacting surface area. As all the studied copolymers fulfilled these requirements, it was possible to determine their polarity indexes and overall polarity (ΣΔ*I*) using this technique. A polarity index ranks adsorbents by the sum of McReynolds’ constant of the test substances. Generally, the higher the polarity index, the more polar the adsorbent. Table 6 presents Kovats’ retention indices for the McReynolds’ test substances determined at 140 °C. The lowest value of ΣΔ*I*, e.g., 441 was observed for BMA-*co*-DVB copolymer. This is a higher value compared to the one of ST-*co*-DVB copolymer determined in analogical way in our previous study [52]. The replacement of DVB by methacrylate comonomers causes a considerable increase in overall polarity. EGDMA and TRIM have the same functional group. As a consequence, the ΣΔ*I* is almost identical for both BMA-*co*-EGDMA and BMA-*co*-TRIM copolymers.

In the last stage of the investigation, the obtained copolymers were subjected to ultraviolet radiation. It turned out that they exhibit fluorescence under UV light (Figure 8). This phenomenon is associated with the presence of anthracene rings in the structure of the porous copolymers.

## 4. Conclusions

The main aim of the presented paper was synthesis and characterization porous, chemically diversified, thermally stable polymeric microspheres. To achieve this goal, BMA was used as functional monomer whereas DVB, EGDMA and TRIM served mainly as crosslinkers. The application of suspension polymerization allowed for the creation of porous, highly crosslinked materials. It was found out that in the case of BMA copolymers, 1.5 wt.% of suspension stabilizer is required to obtain the product in the form of regular microspheres. The porous structure of the obtained microbeads strongly depends on the used comonomer and porogen type. The most developed internal structure possesses BMA-*co*-TRIM_4 copolymer synthesized in the presence of chlorobenzene as a pore-forming diluent. The incorporation of anthracene rings in the copolymers structure as well as high degree of crosslinking contributes to very good thermal stability in oxidative and inert atmosphere. The presence of different functional groups on the surface is responsible for different kind of interaction when the microspheres serve as adsorbents what was evidenced via inverse gas chromatography. What is interesting, the obtained porous copolymers exhibit fluorescence under UV light. They can be potentially applied as precursor for thermally resistant sensors and this direction is going to be the subject of our further investigation.

## Figures and Tables

**Figure 1 materials-16-02610-f001:**
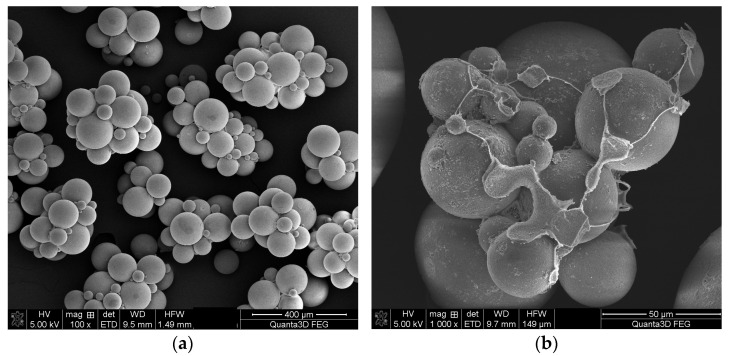
SEM image of BMA-*co*-TRIM_C microspheres synthesized in the presence of 0.5 wt.% (**a**) and 2.5 wt.% (**b**) of PVA.

**Figure 2 materials-16-02610-f002:**
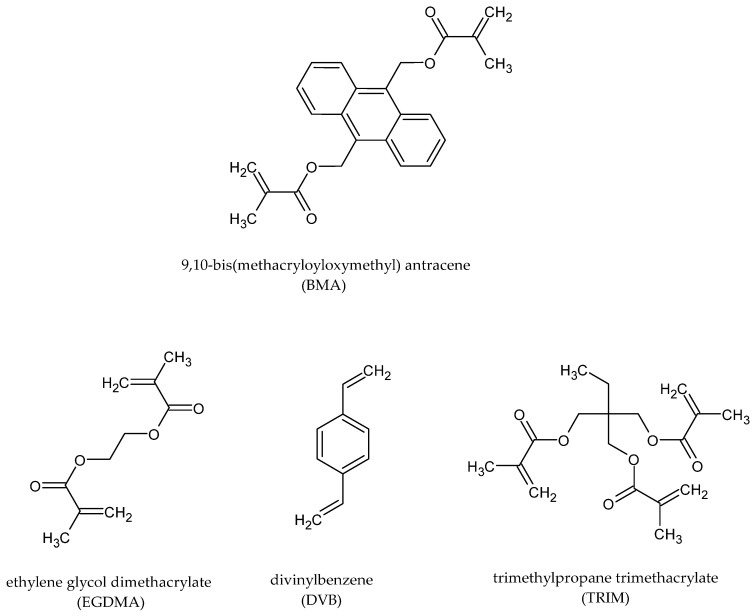
The chemical formulas of the used monomers.

**Figure 3 materials-16-02610-f003:**
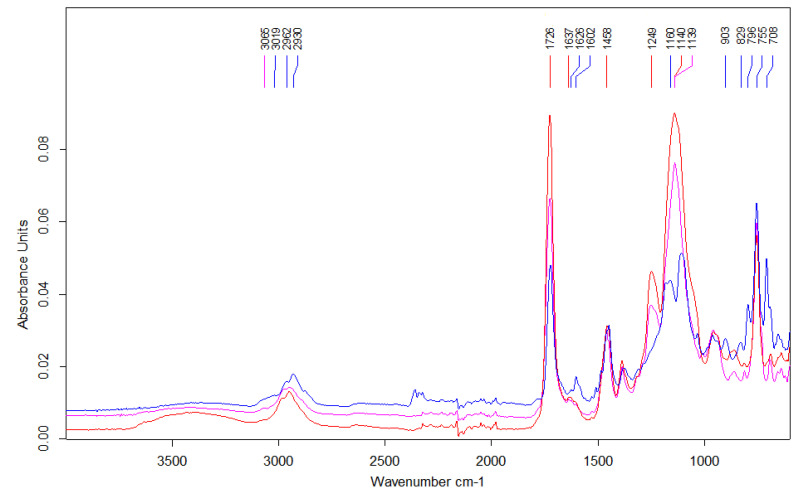
ATR-FTIR spectra of BMA-*co*-DVB (blue line), BMA *co*-EGDMA (pink line) and BMA-*co*-TRIM (red line) copolymers.

**Figure 4 materials-16-02610-f004:**
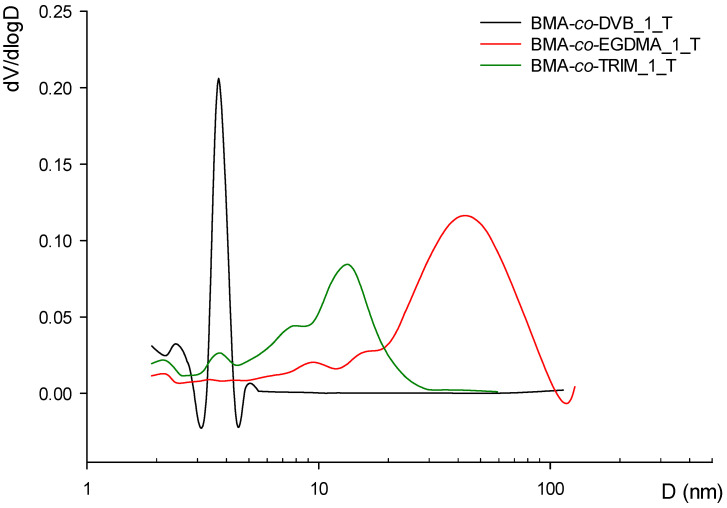
Pore size distribution for selected copolymers.

**Figure 5 materials-16-02610-f005:**
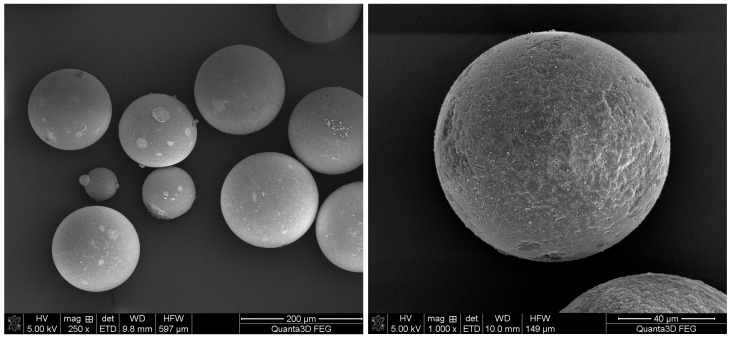
SEM images of the BMA-*co*-DVB_1_T (**left**) and BMA-co-TRIM_1_T (**right**) copolymers.

**Figure 6 materials-16-02610-f006:**
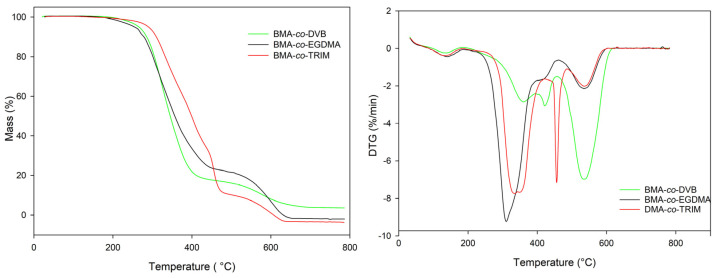
TG (**left**) and DTG (**right**) curves of the investigated copolymers obtained in helium.

**Figure 7 materials-16-02610-f007:**
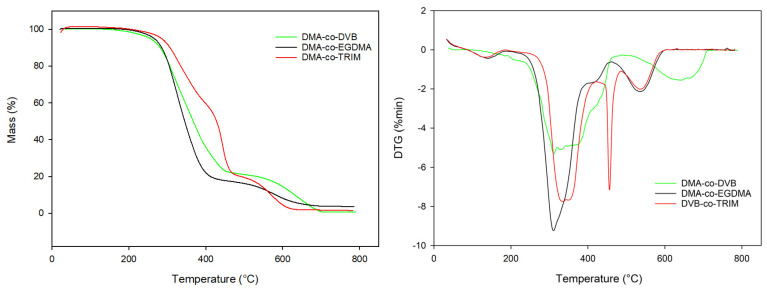
TG (**left**) and DTG (**right**) curves of the investigated copolymers obtained in synthetic air.

**Figure 8 materials-16-02610-f008:**
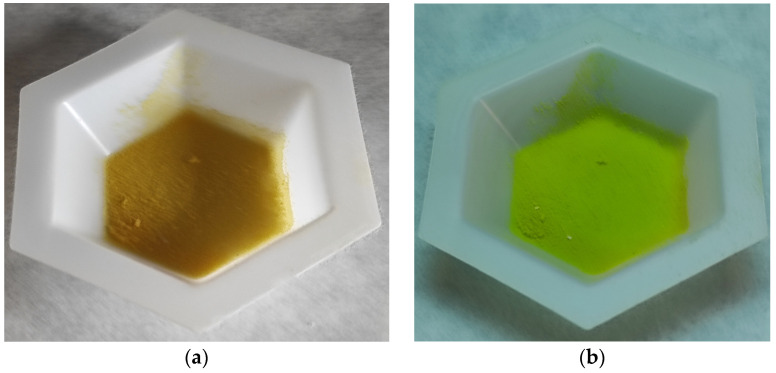
The images of the BMA-*co*-TRIM_4_T copolymer (**a**) before and (**b**) under UV radiation.

**Table 1 materials-16-02610-t001:** The main experimental parameters of the microsphere synthesis.

Copolymer	Monomers		Monomers Ratio	Diluents (mL)	
				Toluene	Chlorobenzene
BMA-*co*-EGDMA_1_T	BMA	EGDMA	1:1	22.5	-
BMA-*co*-EGDMA_1_C		EGDMA	1:1		22.5
BMA-*co*-EGDMA_4_T		EGDMA	1:4	22.5	-
BMA-*co*-EGDMA_4_C		EGDMA	1:4		22.5
BMA-*co*-DVB_1_T		DVB	1:1	22.5	-
BMA-*co*-DVB_1_C		DVB	1:1		22.5
BMA-*co*-DVB_4_T		DVB	1:4	22.5	-
BMA-*co*-DVB_4_C		DVB	1:4		22.5
BMA-*co*-TRIM_1_T		TRIM	1:1	22.5	
BMA-*co*-TRIM_1_C		TRIM	1:1		22.5
BMA-*co*-TRIM_4_T		TRIM	1:4	22.5	
BMA-*co*-TRIM_4_C		TRIM	1:4		22.5

**Table 2 materials-16-02610-t002:** Parameters characterizing porous structure of the copolymeric microspheres.

Copolymer	Specific Surface Area *S_BET_* (m^2^/g)	Pore Volume *V* (cm^3^/g)	Pore Diameter *D_BJH_* (nm)
BMA-*co*-EGDMA_1_T	134.30 ± 0.34	0.1641 ± 0.0012	40
BMA-*co*-EGDMA_1_C	220.49 ± 0.36	0.6299 ± 0.0009	3.9
BMA-*co*-EGDMA_4_T	220.91 ± 0.48	0.5953 ± 0.0021	37
BMA-*co*-EGDMA_4_C	411.94 ± 0.99	0.6410 ± 0.0005	4.1/11 *
BMA-*co*-DVB_1_T	252.32 ± 0.64	0.2028 ± 0.0011	3.5
BMA-*co*-DVB_1_C	384.28 ± 0.14	0.4099 ± 0.0008	4
BMA-*co*-DVB_4_T	358.13 ± 0.57	0.2398 ± 0.0009	2.5
BMA-*co*-DVB_4_C	412.34 ± 0.71	0.4110 ± 0.0015	4.8/17 *
BMA-*co*-TRIM_1_T	265.46 ± 0.42	0.3327 ± 0.0036	3.6/15 *
BMA-*co*-TRIM_1_C	440.34 ± 0.31	0.7309 ± 0.0004	3.5/12 *
BMA-*co*-TRIM_4_T	335.93 ± 0.44	0.4618 ± 0.0022	10
BMA-*co*-TRIM_4_C	472.36 ± 0.29	0.7498 ± 0.0007	8.5

* bimodal pore size distribution.

**Table 3 materials-16-02610-t003:** The statistical data of particle size distribution.

Copolymeric Microsphere	*D*(0.1) (µm)	*D*(0.5) (µm)	*D*(0.9) (µm)	Span
BMA-*co*-EGDMA_1_T	94	132	164	0.530
BMA-*co*-EGDMA_1_C	98	141	179	0.574
BMA-*co*-EGDMA_4_T	89	138	171	0.592
BMA-*co*-EGDMA_4_C	101	147	184	0.565
BMA-*co*-DVB_1_T	68	95	101	0.347
BMA-*co*-DVB_1_C	70	97	128	0.598
BMA-*co*-DVB_4_T	64	92	114	0.543
BMA-*co*-DVB_4_C	73	100	136	0.624
BMA-*co*-TRIM_1_T	80	162	181	0.623
BMA-*co*-TRIM_1_C	98	174	201	0.592
BMA-*co*-TRIM_4_T	76	168	196	0.714
BMA-*co*-TRIM_4_C	102	183	217	0.628

**Table 4 materials-16-02610-t004:** The characteristic temperatures determined on the basis of TG and DTG data (helium atmosphere).

Copolymeric Microspheres	*T*_5%_ (°C)	*T*_20%_ (°C)	*T*_50%_ (°C)	*T*_max1_ (°C)	*T*_max2_ (°C)	*T*_max2_ (°C)	FDT
BMA-*co*-DVB_1	256	307	367	335	425	646	720
BMA-*co*-DVB_4	261	312	381	343	431	658	738
BMA-*co*-EGDMA_1	254	301	356	320	417	571	673
BMA-*co*-EGDMA_4	241	286	341	311	402	562	664
BMA-*co*-TRIM_1	304	346	434	325	442	571	680
BMA-*co*-TRIM_4	309	358	452	336	456	583	692

**Table 5 materials-16-02610-t005:** The characteristic temperatures determined on the basis of TG and DTG (synthetic air atmosphere).

Copolymeric Microspheres	*T*_5%_ (°C)	*T*_20%_ (°C)	*T*_50%_ (°C)	*T*_max1_ (°C)	*T*_max2_ (°C)	*T*_max3_ (°C)	FDT
BMA-*co*-DVB_1	251	316	367	360	421	626	701
BMA-*co*-DVB_4	258	318	374	368	429	641	718
BMA-*co*-EGDMA_1	250	303	339	310	414	536	591
BMA-*co*-EGDMA_4	243	298	327	302	408	528	583
BMA-*co*-TRIM_1	301	327	369	335	456	536	596
BMA-*co*-TRIM_4	307	331	374	339	461	542	601

**Table 6 materials-16-02610-t006:** Kovats’ retention indices for the McReynolds’ test substances and overall polarity (ΣΔ*I*) for the porous copolymers.

Copolymeric Microsphere	Kovats’ Retention Indices			McReynolds’ Constants			
	*I_x_* (Benzene)	*I_y_* (Butan-1-ol)	*I_z_* (Pentan-2-one)	Δ*I_x_*	Δ*I_y_*	Δ*I_z_*	ΣΔ*I*
BMA-*co*-DVB_1	649	698	718	75	210	156	441
BMA-*co*-EGDMA_1	673	753	749	99	264	184	547
BMA-*co*-TRIM_1	669	748	759	95	259	194	548
GTCB	574	489	565	-	-	-	-

## Data Availability

The data presented in this study are available on request from the corresponding authors.

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
