# Peer review of "Porous Polymers Based on 9,10-Bis(methacryloyloxymethyl)anthracene—Towards Synthesis and Characterization"

_materials, 2023, doi:10.3390/ma16072610_

Round 1

Reviewer 1 Report

In the research article titled “Porous polymers based on 9,10-bis(methacryloyloxymethyl) anthracene - towards synthesis and characterization” by Maciejewska et al., authors have presented the good analysis, but there are many issues which I think should be addressed. The highlighted issues are as follow;

1.       In abstract, authors have measured many properties of the materials but highlighted only the specific surface area in the range of 134-472 16 m2/g, which makes the work very weak, I suggest authors should highlight all the aspects they have measured in the abstract to make it appealing for the readers.

2.      One important aspect, there should be at least one precise sentence in the abstract regarding the practical application of this material in society. Second, are these measured characteristics are enough for that applications?

3.      Porous polymers are interesting materials but there should be some other measurements to make the article strong, like its composite for energy storage applications or some other.

4.      Please plot the ATR-FTIR spectra in origin along with the indexing of the peaks.

Author Response

Thank you for the valuable comments. We greatly appreciate the Reviewer’s insight into the manuscript and constructive feedback. The revised version of our manuscript takes into account the recommendations of all the Reviewers. The significant corrections are highlighted in yellow.

  1. The Abstract has been changed. Most of the measured aspects are highlighted. The potential practical application of the synthesized materials are mentioned.
  2. In addition to the previously determined properties, the behavior of the studied copolymers under UV light was described.
  3. ATR spectra in origin along with the indexing of the peaks are provided.

Reviewer 2 Report

Maciejewska et al. described a study on preparation and characterization of permanently porous 8 and thermally stable polymeric microspheres. Although the manuscript is well written, proper contexts are missing in the manuscript. Below are points of attention from this reviewer:

Point 1: This reviewer thinks the Abstract needs to be revised. It should contain 1-2 lines of background, 2-3 lines of manuscript objective and methods, followed by findings and lastly the significance of this work and/or future directions. Currently the abstract looks more like conclusions and the background and significance/future directions are missing.

Point 2: It is important to distinguish the necessity and uniqueness of this manuscript when there are several papers and reviews being published on this popular topic. Authors are advised to clarify the motivation and innovation (differences between this paper and others) of this article compared with other papers, and add them accordingly to Section 1.

Point 3: Authors are advised to perform error analysis for all numerical results.

Point 4: Noises in Figure 4 and 7 should be explained for failure/error analysis.

Point 5: In the Conclusion, authors should restate the thesis and show how it has been developed through the body of the paper, ending with future directions. Briefly summarize the key arguments made in the body, showing how each of them contributes to proving your thesis. Authors are advised to rewrite this section and add future directions, failure causes analysis, etc.

Point 6: English must be improved for this manuscript.

Author Response

Thank you for the valuable comments. We greatly appreciate the Reviewer’s insight into the manuscript and constructive feedback. The revised version of our manuscript takes into account the recommendations of all the Reviewers. The significant corrections are highlighted in yellow.

Answers to comments  in the Review 2:

  1. The Abstract has been revised according to the substantive suggestions of the Reviewer.
  2. The authors’ motivation and paper innovation is accentuated in Section 1.
  3. The accessible error data is provided.
  4. The noises in Figure7 are associated with the high sensitivity of the thermal analyzer. They occurred at the end of measurements and did not have relevant impact on the acquired results. In the case of Figure 4 they are connected with the type of polynomial used in the ASAP data processing.
  5. The Conclusion has been rewritten according to the Reviewer’s guidelines.
  6. English has been carefully checked and corrected throughout the whole manuscript.

Round 2

Reviewer 1 Report

Authors have revised the manuscript very well according to my suggestion. I am satisfied with the revised version. I think now this article can be publish in the materials.

Reviewer 2 Report

All comments have been addressed so this reviewer suggests acceptance for publication.